# Nanobody-triggered lockdown of VSRs reveals ligand reloading in the Golgi

Simone Frühholz[1], Florian Fäßler [1], Üner Kolukisaoglu[1] & Peter Pimpl [1,2]

Protein degradation in lytic compartments is crucial for eukaryotic cells. At the heart of this process, vacuolar sorting receptors (VSRs) bind soluble hydrolases in the secretory pathway and release them into the vacuolar route. Sorting efficiency is suggested to result from receptor recycling. However, how and to where plant VSRs recycle remains controversial. Here we present a nanobody–epitope interaction-based protein labeling and tracking approach to dissect their anterograde and retrograde transport routes in vivo. We simultaneously employ two different nanobody–epitope pairs: one for the location-specific post-translational fluorescence labeling of receptors and the other pair to trigger their compartment-specific lockdown via an endocytosed dual-epitope linker protein. We demonstrate VSR recycling from the TGN/EE, thereby identifying the *cis*-Golgi as the recycling target and show that recycled VSRs reload ligands. This is evidence that bidirectional VSR-mediated sorting of vacuolar proteins exists and occurs between the Golgi and the TGN/EE.

[1] Center for Plant Molecular Biology (ZMBP), University of Tübingen, Auf der Morgenstelle 32, 72076 Tübingen, Germany. [2] SUSTech-PKU Institute of Plant and Food Science (IPFS), Department of Biology, Southern University of Science and Technology (SUSTech), 1088 Xueyuan Rd, Shenzhen 518055, China. Correspondence and requests for materials should be addressed to P.P. (email: pimpl@sustc.edu.cn)

Degradation in lytic compartments is a hallmark of eukaryotic cells. It allows for rapid modulations of compartmental protein and lipid compositions as responses to cellular communication or environmental cues[1–4]. This necessitates constant supply of vacuoles/lysosomes with acid hydrolysis by the action of sorting receptors[5]. Despite its significance for viability and development, the core mechanism of vacuolar sorting receptor (VSR)-mediated protein transport and its implementation in the plant endomembrane system is still controversial[5,6].

The concept of receptor-mediated protein transport dates back to the discovery of the low-density lipoprotein receptor and the cation independent (CI)-mannose 6-phosphate receptor (MPR) for lysosomal sorting in mammals[7–9]. They bind ligands either at the cell surface or the TGN and transport them to endosomes, where ligands are released due to low compartmental pH[8,10]. The key to the efficiency of this transport however, is the continuous recycling of receptors after ligand release, allowing receptors to go through hundreds of transport-cycles during their lifespan[7,8,11–13].

The recycling route of MPRs was most elegantly mapped biochemically, by assaying for Golgi cisternae-specific glycan processing after receptor labeling with [3H] galactose at the cell surface by using exogenous galyctosyltransferases[14]. However, endogenous VSRs do not localize to the cell surface and are thus not amendable to exogenously applied modifying enzymes to decipher their function or to trace their transport route in vivo.

VSRs are type I transmembrane proteins and bind ligands via a luminal ligand-binding domain (LBD), whereas their cytosolic tail carries the sorting information for their own transportation[15–23]. They were originally proposed to transport ligands into prevacuoles, nowadays referred to as multivesicular bodies/late endosomes (MVBs/LEs)[16,19,21,24–26]. However, we have recently demonstrated that VSRs bind ligands in the early secretory pathway and instead release them in the trans-Golgi network (TGN)[27], the early endosome (EE) of plants (TGN/EE)[28,29]. This raised the fundamental questions as to how to where VSRs recycle after ligand release. To address this, we have devised a strategy that utilizes the in vivo interaction of two different antibody-epitope pairs. This allows (a) for the location-specific green fluorescent protein (GFP)-labeling of VSRs in the TGN/EE and (b) for the tracking and lockdown of such labeled VSRs in upstream compartments, upon retrograde recycling. For this, we have translationally fused a variable domain of a lama (Lama paco) heavy-chain antibody (V$_H$H)[27,30], termed nanobody (Nb), that was raised against GFP (Nb$_G$)[27,31] to a VSR (Nb$_G$-VSR). The other Nb, which was raised against α-synuclein (Nb$_S$)[32], was fused to compartment-specific membrane marker proteins. Finally, we have designed a dual-epitope linker protein, which contains the epitopes of both nanobodies and therefore allows for both, specific GFP-labeling of the Nb$_G$-tagged VSR via the GFP domain and attachment to Nb$_S$-tagged compartmental marker proteins via the α-synuclein (SYN) epitope. Labeling of Nb$_G$-VSR in the lumen of the TGN/EE is achieved by incubation of Nb$_G$ and Nb$_S$ fusion protein-expressing cells with the dual-epitope linker protein GFP-SYN, which is endocytosed and delivered to the TGN/EE. Using this approach, we have traced GFP-labeled VSRs from the TGN/EE back to the cis-Golgi, where we demonstrate their ligand-binding capability. Altogether, these data demonstrate the cycling of VSRs between the Golgi stack and the TGN/EE.

## Results

### Post-translational GFP-labeling via endocytosed GFP. The challenge when analyzing bidirectional protein transport of sorting receptors in live-cell imaging studies is to differentiate between anterograde and retrograde transported receptors under steady state conditions. This is particularly true when translational fusions between receptors and fluorescent proteins are used. Here, fluorescent signals become detectable immediately after synthesis and protein folding in the ER and they persist throughout the lifespan of the molecule. Consequently, the localization of the receptor does not provide any information on its transport direction or ligand status (Fig. 1a). An analysis of receptor recycling therefore demands strategies that allow for the specific tracing of those VSRs that have released their ligands in the TGN/EE[27] and are about to be recycled. This requirement is fulfilled if a post-translational labeling strategy is used where signals of the labeled VSRs become first detectable in the TGN/EE (Fig. 1b). To achieve this, we have devised an approach that accounts for both, the target-specificity of the labeling and the intracellular location where the labeling occurs. For this, we have employed a GFP-binding nanobody (Nb$_G$)[27,31] that is translationally fused to the VSR and we deliver its epitope GFP as the fluorescent labeling agent to the TGN/EE via endocytosis. We produced the labeling GFP as a secretory protein in another population of tobacco mesophyll protoplasts. The resulting GFP-containing culture medium is then used for labeling of the cell population that expresses the Nb$_G$-tagged VSRs. This strict separation between cells that produce the labeling agent GFP and cells that are used for the labeling ensures that no newly synthesized VSR on its anterograde route is labeled prior to reaching the TGN/EE.

To develop a compartment-specific post-translational GFP-labeling strategy, we firstly decided to employ the established marker proteins α-mannosidase 1 (Man1) for the cis-Golgi, sialyltransferase (ST) for the trans-Golgi, SYNTAXIN OF PLANTS (SYP) 61 for the TGN/EE, and the luminal ligand-binding domain-deprived (ΔLBD) binding protein 80 kDa (BP80) from Pisum sativum for MVB/LE in coexpression experiments to discriminate between the various punctate signals (Supplementary Fig. 1). Next, we generated and tested red fluorescent protein (RFP)-tagged Nb$_G$ fusion proteins of these markers in tobacco mesophyll protoplasts for post-translational labeling in the TGN/EE (SYP61-RFP-Nb$_G$), the MVB/LE (Nb$_G$-RFP-BP80ΔLBD), the trans-Golgi (ST-RFP-Nb$_G$), the cis-Golgi (Man1-RFP-Nb$_G$) and the ER (Nb$_G$-RFP-Calnexin (CNX)). After transfection with the respective marker construct, we incubated the cells in GFP-containing culture medium for the endocytic uptake of GFP (endocyt GFP) (Fig. 1c–g). Confocal laser-scanning microscopy (CLSM) demonstrated that the endocytosed GFP was trapped by the Nb$_G$-tagged markers SYP61-RFP-Nb$_G$ and Nb$_G$-RFP-BP80ΔLBD in the TGN/EE and downstream in the MVB/LE, respectively (Fig. 1c, d). In sharp contrast, labeling of the markers in compartments upstream of the TGN/EE like the trans-/cis-Golgi (ST-RFP-Nb$_G$, Man1-RFP-Nb$_G$) or the ER (Nb$_G$-RFP-CNX) with endocytosed GFP was never observed (Fig. 1e–g). However, post-translational GFP-labeling based on Nb$_G$-epitope interaction is also possible in these compartments, if the labeling agent GFP is coexpressed as a secretory protein (Sec-GFP) with the respective Nb$_G$-fusion proteins (Fig. 1h–j). This shows that post-translational GFP-labeling via Nb$_G$-epitope interaction is applicable to Nb$_G$-tagged proteins in all compartments and furthermore demonstrates that endocytosed GFP alone does not reach locations upstream of the TGN/EE like the cis-/trans-Golgi and the ER. Consequently, this also demonstrates that none of the Nb$_G$-tagged markers that reside in the ER or the Golgi apparatus, ever reach or cycle through the TGN/EE in order to reach their respective steady state distribution.

### Post-translationally labeled VSRs localize to the TGN/EE. In the next step, we applied this post-translational GFP-labeling

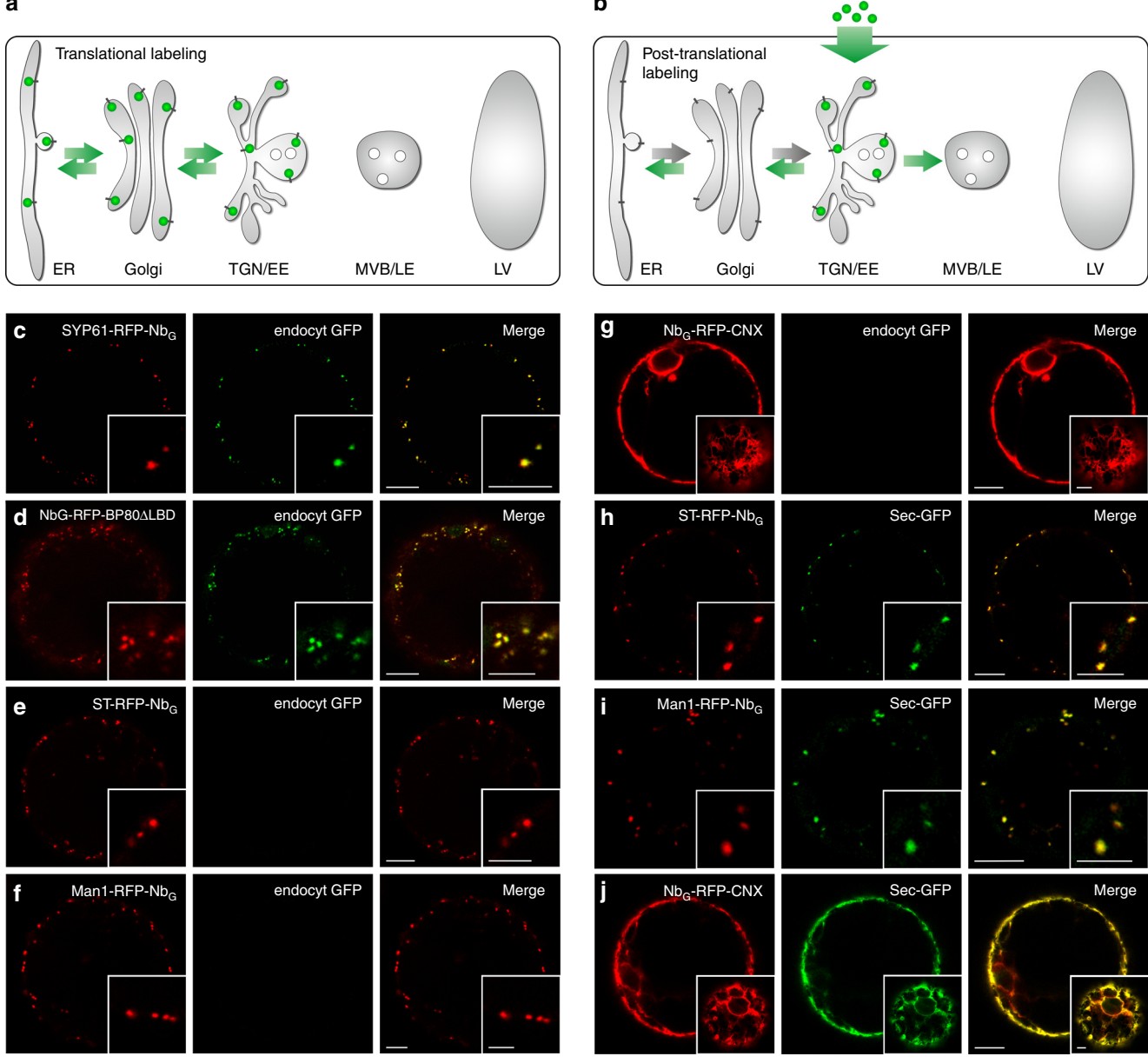

**Fig. 1** Post-translational GFP-labeling via nanobody–epitope interaction. **a** Translational GFP-labeling of VSRs. **b** Post-translational GFP-labeling of a $Nb_G$-tagged VSR in the TGN/EE by endocytosed GFP. **c**–**g** Post-translational GFP-labeling of compartment-specific $Nb_G$-tagged red fluorescent membrane anchors (red) by endocytosed GFP (green) in **c** the TGN/EE and **d** the MVB/LE. Endocytosed GFP does not reach **e** the *trans*-Golgi, **f** the *cis*-Golgi nor **g** the ER. **h**–**j** Post-translational GFP-labeling by coexpression of secreted (Sec)-GFP (green) and $Nb_G$-tagged red fluorescent membrane anchors (red) for **h** the *trans*-Golgi, **i** the *cis*-Golgi and **j** the ER. Insets in **g**, **j** show cortical sections, others show magnifications. Scale bars 10 μm, insets 5 μm

protocol to VSRs (Fig. 2a, b). To better judge the labeling efficiency, we tagged a fluorescent VSR[33] with the $Nb_G$ ($Nb_G$-RFP-VSR) and performed post-translational GFP-labeling (Fig. 2b). CLSM-based colocalization reveals almost perfectly matching punctate signals of the red $Nb_G$-RFP-VSR and the green signals from the endocytosed GFP (Fig. 2c, d), demonstrating a high degree of labeling efficiency. However, since $Nb_G$-RFP-VSR can acquire the labeling GFP only in the TGN/EE, this high degree of colocalization furthermore suggests that under steady state conditions almost all of the $Nb_G$-RFP-VSR molecules had already reached the TGN/EE, at least once.

We have recently demonstrated that VSRs bind their ligands in the ER, in the *cis*- and *trans*-Golgi, but release their ligands in the TGN/EE[27]. Therefore, we hypothesized that the ligand-free

receptors that were post-translationally labeled with endocytosed GFP in the TGN/EE would recycle to an upstream compartment for ligand reloading. In such a scenario, we would then expect to detect a population of labeled VSRs in a compartment upstream of the TGN/EE. To precisely define the VSR localizations we next post-translationally labeled non-fluorescent $Nb_G$-tagged VSRs ($Nb_G$-VSR) with endocytosed GFP and tested for colocalization with established red fluorescent compartmental markers (Fig. 2e, f) for the TGN/EE (RFP-SYP61), the MVB/LE and vacuole (Aleu-RFP), *trans*- and *cis*-Golgi (ST-RFP and Man1-RFP, respectively) and for the ER (RFP-CNX) in coexpression experiments (Fig. 2g–m). The post-translationally labeled $Nb_G$-VSRs colocalized with the TGN/EE marker (Fig. 2g). Surprisingly, the post-translationally labeled $Nb_G$-VSRs neither colocalized

with the MVB/LE and vacuole marker Aleu-RFP (Fig. 2h, i) nor with markers for upstream compartments like the *trans-* and *cis-*Golgi or the ER (Fig. 2j–m). This steady state localization of $Nb_G$-VSR at the TGN/EE rather than at the MVB/LE, as is commonly assumed, is not restricted to post-translationally labeled $Nb_G$-VSRs, it is also seen in control experiments using the fluorescent full-length receptor fusion protein $Nb_G$-RFP-VSR (Supplementary Fig. 2). The differential localization of these full-length VSRs

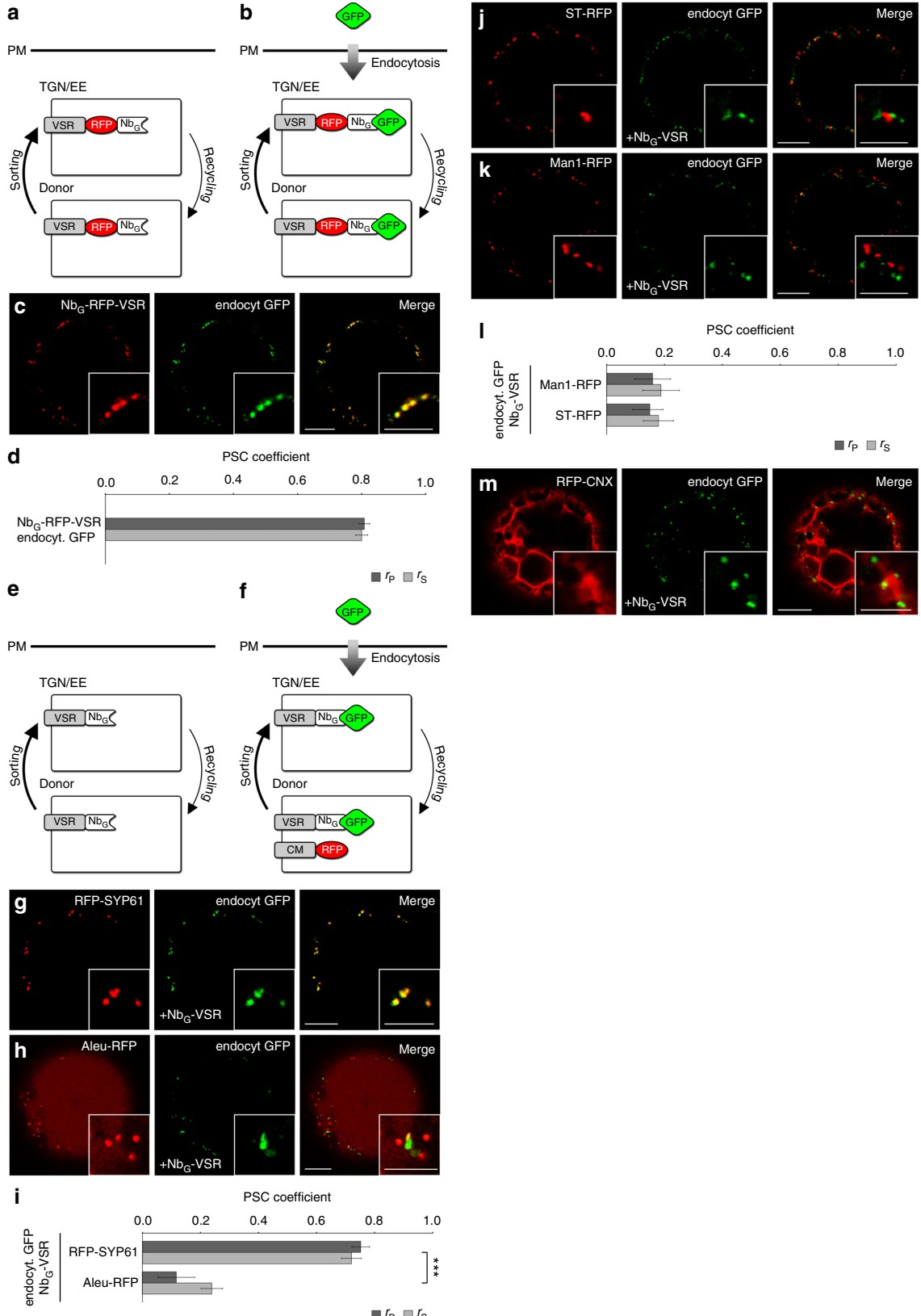

and LBD-lacking MVB/LE markers of the RFP/GFP-BP80ΔLBD type, therefore suggests that the presence of the LBD is required for both, the ligand-binding capability and for the correct transport of the receptor.

**Nanobody-triggered lockdown of recycled VSRs.** One possible explanation for the TGN/EE-localization of VSRs under steady state conditions is that VSRs do not recycle to reload ligands. Such a one-way transport mode was suggested for members of the receptor homology region-transmembrane domain-RING-H2 (RMR) receptor family, which sort proteins to the protein storage vacuole[34]. However, considering that the TGN/EE is expected to be the recycling point of a bidirectional transport system[27,35], the TGN/EE-localization of cycling VSRs may indicate that ante-rograde transport is faster than the subsequent recycling step. To test for his hypothesis, we have devised a strategy that allows for the specific detection of recycled receptors in compartments upstream of the TGN/EE by blocking their further export and onward forwarding upon completion of recycling. For this, we have combined the nanobody-mediated post-translational label-ing of recycling VSRs in the TGN/EE with an approach for compartment-specific lockdown of these labeled VSRs via a nanobody-epitope interaction that is triggered by a second nanobody–epitope pair (Fig. 3a–c).

Hereto, we translationally fused a nanobody that is directed against the mammalian α-synuclein (Nb$_S$) to red fluorescent compartment-specific markers (CM-RFP-Nb$_S$) and we fused its corresponding epitope termed SYN, which is a sequence of 23 amino acids, to GFP (GFP-SYN). Endocytic uptake of this dual-epitope linker as the labeling agent by cells coexpressing Nb$_G$-VSRs and Nb$_S$-tagged compartmental markers was then expected to firstly label Nb$_G$-VSRs in the TGN/EE and then to trigger an in vivo crosslink between the SYN epitope of the GFP-SYN-labeled VSR and the Nb$_S$-tagged compartmental marker in the compartmental lumen.

This complex strategy required that we first test whether Nb$_S$ interacts with the SYN epitope in the lumen of secretory pathway compartments. To this end, we developed an assay for analyzing protein-protein interaction in vivo. This assay is based on the simultaneous use of a quantifiable soluble secretory reporter with a soluble vacuolar protein, each of which carries either the nanobody or the epitope, respectively. In this approach, the interaction occurring between the Nb$_S$ and the epitope triggers the attachment of the vacuolar sorting signal to the secretory reporter and consequently, its transport to the lytic vacuole via the vacuolar sorting machinery (Fig. 3d, e).

We therefore tagged the secretory reporter α-amylase from barley (*Hordeum vulgare*)[36] with the SYN epitope (amylase-SYN) and employed the vacuolar reporter Aleu-RFP as a Nb$_S$-fusion protein (Aleu-RFP-Nb$_S$). Quantitative transport analysis of the secretory amylase-SYN in tobacco mesophyll protoplasts shows that its secretion is drastically reduced by the coexpressed vacuolar Aleu-RFP-Nb$_S$ (Fig. 3f), suggesting an interaction

between the Nb$_S$ and the SYN epitope. In the next step, we tested the functionality of Nb$_S$ in the context of the compartment-specific membrane anchors for the ER, *cis*- and *trans*-Golgi and the TGN/EE. For this, we fused the Nb$_S$ to RFP-CNX (Nb$_S$-RFP-CNX), Man1-RFP (Man1-RFP-Nb$_S$), ST-RFP (ST-RFP-Nb$_S$) and SYP61-RFP (SYP61-RFP-Nb$_S$) and verified firstly their correct location in colocalization experiments with their respective GFP-tagged counterpart (Supplementary Fig. 3). Second, we tested their ability to bind the dual-epitope linker GFP-SYN (Fig. 3g–j). To do this, we immunoprecipitated the above-mentioned marker proteins and their Nb$_S$-tagged pendants, the anchors, with RFP antibodies in bead-binding assays and subjected all of them to the GFP-SYN-containing culture medium from GFP-SYN-secreting protoplasts. The immunoblot analysis of the precipitates revealed that all of the Nb$_S$-tagged anchors co-immunoprecipitated the SYN epitope-tagged GFP, whilst this molecule was absent in precipitates from markers lacking the Nb$_S$. To rule out that on the other side the SYN epitope from GFP-SYN perturbs the interaction between the GFP epitope and Nb$_G$, we performed comparative co-immunoprecipitation experiments using bead-bound Nb$_G$-VSR with either secreted GFP or secreted GFP-SYN, to show that the Nb$_G$-VSR binds GFP and GFP-SYN to comparable levels (Fig. 3k).

Finally, we performed colocalization experiments of GFP-SYN-labeled Nb$_G$-VSRs with the markers for the TGN/EE, *trans*- and *cis*-Golgi and the ER, showing that the labeling of Nb$_G$-VSR with GFP-SYN does not alter the localization of the labeled VSR (Supplementary Fig. 4, compare to Fig. 2).

Altogether, these results show that this second Nb$_S$-SYN nanobody–epitope pair is suitable for triggering a stable linkage between proteins, both in vitro and in vivo. The results also demonstrate that each epitope of GFP-SYN is accessible for Nb$_G$ or Nb$_S$ interaction.

**The *cis*-Golgi is the target of the VSR recycling route.** To apply the strategy for nanobody-triggered lockdown to the analysis of VSR recycling, we have subjected cells that coexpress Nb$_G$-tagged VSRs with the above-mentioned Nb$_S$-tagged anchors to post-translational VSR-labeling using the endocytosed dual-epitope linker GFP-SYN (Fig. 4). First labeling of Nb$_G$-VSRs in cells coexpressing the TGN/EE anchor SYP61-RFP-Nb$_S$ resulted in almost perfect colocalization of both signals (Fig. 4a), as was seen before when the non-tagged TGN/EE marker was used (Fig. 4b). This suggested that the endocytosed Nb$_G$-VSR-labeling agent GFP-SYN does not generally perturb membrane trafficking events in the presence of the Nb$_S$-tagged membrane anchor. In the next step, we subjected cells that coexpressed the anchors for the upstream compartments to this procedure. Here, the locali-zation of the GFP-SYN-labeled Nb$_G$-VSRs shifted drastically and now colocalized with the *trans*-Golgi anchor ST-RFP-Nb$_S$ (Fig. 4c, d, compare to 4b). Likewise, localization of GFP-SYN-labeled Nb$_G$-VSRs shifted strongly towards the *cis*-Golgi when the anchor Man1-RFP-Nb$_S$ was used for the lockdown of the labeled

---

**Fig. 2** Localization of post-translationally labeled Nb$_G$-tagged VSRs. **a** Cycling Nb$_G$-tagged red fluorescent VSRs are **b** post-translationally labeled by endocytosed GFP. **c** GFP-labeled red fluorescent Nb$_G$-tagged VSRs. **d** Pearson's ($r_P$) and Spearman's ($r_S$) correlation (PSC) coefficients of Nb$_G$-RFP-VSRs and labeling GFP. Data are presented as average ± s.e.m. of 10 individual cells. The graph shows a representative sample of two independent experiments. **e, f** Colocalization of post-translationally GFP-labeled non-fluorescent cycling Nb$_G$-tagged VSRs (Nb$_G$-VSR) with red fluorescent compartmental markers (CM) for **g, i** the TGN/EE, **h, i** MVBs/LEs and vacuole, **j, l** the *trans*− and **k, l** *cis*-Golgi and **m** the ER. **i, l** PSC coefficients of the labeled Nb$_G$-VSR and coexpressed markers RFP-SYP61, Aleu-RFP, Man1-RFP and ST-RFP are calculated and presented as in **d**. Graphs show a representative sample of two independent experiments. Labeled Nb$_G$-VSRs colocalize with the TGN/EE marker but not with markers for MVB/LE and vacuole and the *cis*-/*trans*-Golgi. **i** Significance was calculated using unpaired, two tailed *t*-test ($n = 10$, ***$P < 0.001$). Scale bars 10 μm, insets 5 μm. Insets show magnifications

NbG-VSR (Fig. 4e, f, compare to 4b). The colocalization of GFP-SYN-labeled VSRs and the NbS-tagged anchors for the *trans*- and *cis*-Golgi strictly depends on the presence of the second nanobody–epitope pair and was never observed when marker pendants without the NbS-tag were used (Fig. 4d, f, compare to

Supplementary Figs. 4 and 5). This suggests, that the VSRs did indeed recycle from the TGN/EE to the Golgi stack. To rule out that the GFP-SYN triggered crosslink between NbG-VSR and ST-RFP-NbS or Man1-RFP-NbS altered the Golgi localization of the anchors in these experiments, we used the fungal toxin brefeldin A (BFA) as a diagnostic tool to confirm the Golgi localization of both anchors. In tobacco, BFA causes a fusion between Golgi stacks and the ER[37] thereby triggering a shift of signals from Golgi anchors to the ER. After BFA-treatment, the punctate signals from crosslinked GFP-SYN-labeled VSR-*cis*- and *trans*-Golgi cisternal anchors became detectable in the nuclear envelope (Fig. 4g, h). This demonstrated that the lockdown did not alter the localization of the Golgi anchors. In sharp contrast, a colocalization between GFP-SYN-labeled NbG-VSRs and the ER anchor NbS-RFP-CNX was never observed (Fig. 4i). This, however, indicates that VSRs do not recycle to upstream compartments further than the *cis*-Golgi.

**Recycled VSRs reload ligands in the *cis*-Golgi.** We have previously used the soluble model ligand Aleu-RFP together with a soluble NbG-tagged LBD of a VSR that was anchored to a GFP-tagged membrane marker by nanobody–epitope interaction. There, ligand binding to the anchored LBD in a given compartment was visualized through coaccummulation/colocalization of the otherwise passing ligands[27]. We have now extended this visualization concept to the analysis of the ligand-binding capabilities of recycled full-length VSRs in the Golgi (Fig. 5a–d).

Hereto, we performed a *cis*-Golgi-specific dual-epitope triggered VSR lockdown in cells, coexpressing the vacuolar reporter Aleu-RFP, NbG-VSR, Man1-NbS, which is used for the lockdown and Man1-blue fluorescent protein (BFP)2, which serves as neutral marker to verify the localization (Fig. 5e). The analysis clearly shows the triple-overlap of the fluorescence signals from the vacuolar reporter Aleu-RFP with the recycled GFP-SYN-labeled NbG-tagged VSR and the *cis*-Golgi marker Man1-BFP2, demonstrating the interaction between the recycled VSRs and the ligand in the *cis*-Golgi. The same was also seen when the VSR lockdown was performed in the *trans*-Golgi by using ST-NbS (Supplementary Fig. 6a). In sharp contrast, no colocalization between VSRs and ligands are seen in controls without the NbS-SYN triggered VSR lockdown: neither in the absence of the NbS-

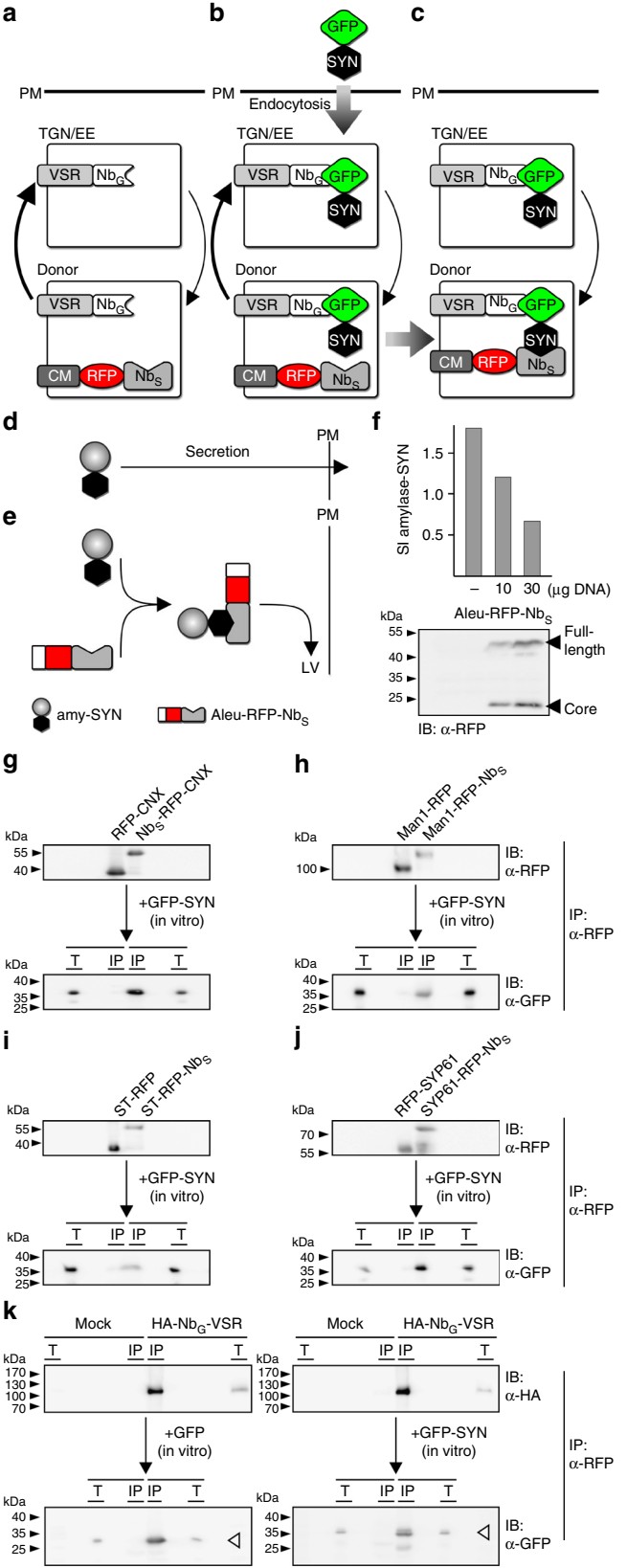

**Fig. 3** Nanobody-triggered lockdown of recycled VSRs. **a** Coexpression of NbG-VSRs with red fluorescent NbS-tagged compartmental markers (anchors) and **b** post-translational labeling with the dual-epitope GFP-SYN in the TGN/EE to **c** anchor VSRs upon recycling. **d, e** NbS-SYN epitope interaction occurs in the endomembrane system. **d** SYN epitope-tagged secreted amylase (amy-SYN) is **e** rerouted to the lytic vacuole (LV) upon Aleu-RFP-NbS triggers attachment of the vacuolar sorting signal (Aleu). **f** Coexpression of amy-SYN with different amounts of Aleu-RFP-NbS. Upper panel: secretion index (SI); lower panel: corresponding immunoblot (α-RFP). **g–j** Co-immunoprecipitations revealing NbS-SYN epitope interaction. RFP-tagged markers and anchors for **g** ER, **h** *cis*-Golgi, **i** *trans*-Golgi and **j** TGN/EE were immunoprecipitated (IP, α-RFP), incubated with GFP-SYN and immunoblotted (IB). Total extracts (T) and immunoprecipitates (IP) were probed to detect markers, anchors (α-RFP) and co-precipitated GFP-SYN (α-GFP). **k** Co-immunoprecipitation revealing NbG-GFP epitope interaction. Expressed NbG-VSRs or samples from mock-transfected cells were immunoprecipitated (IP, α-HA), incubated with GFP or GFP-SYN and immunoblotted (IB). Total extracts (T) and immunoprecipitates (IP) were probed to detect VSRs (α-HA) and co-precipitated GFP (white arrowhead) or GFP-SYN (black arrowhead) (α-GFP), respectively

tagged anchor (Fig. 5f, Supplementary Fig. 6b) nor in the absence of the SYN epitope, when GFP is used for the labeling instead of GFP-SYN (Fig. 5g, Supplementary Fig. 6c). This was to be expected, since "free" labeled VSRs localize to the TGN/EE in these controls (compare to Fig. 2g, i and Supplementary Fig. 4a), a compartment that does not provide ligand-binding conditions[27].

## Discussion

Being only about 125 amino acids long, nanobodies are the smallest entities, capable of specific antigen recognition and binding[38]. Nanobodies are therefore ideally suited for the generation of genetically encoded molecular tools for the identification, localization and manipulation of protein function in living cells for basic research and applied sciences[39,40].

We have previously generated VSR sensors for a compartment-specific analysis of VSR-ligand interactions[27]. They self-assemble from soluble VSR_LBD-Nb$_G$ fusion proteins and GFP-tagged compartment-specific membrane anchors. Using this approach, we have demonstrated that VSRs bind ligands in the ER, the cis- and the trans-Golgi and ultimately release ligands in the TGN/EE[27], thereby opening the question about the fate of VSRs after this step. The analysis of bidirectional receptor transport and receptor recycling in particular, however, is technically most challenging in living cells. It requires molecular tools that permit the strict differentiation between VSRs from the anterograde and the retrograde trafficking route.

To overcome these constraints, we have taken advantage of the TGN as also being the EE by incubating Nb$_G$-VSR-expressing cells with exogenously applied protoplast-secreted GFP to trigger compartment-specific labeling of VSRs in the TGN/EE by its endocytic uptake. This ensures labeling of only those VSRs that have reached the recycling point, whereas newly synthesized VSRs from the anterograde route remain invisible. Most interesting for future application however is the simultaneous use of two different Nb-epitope pairs in vivo. This allows for triggering a protein-specific lockdown of recycled Nb$_G$-VSRs at Nb$_S$-tagged membrane proteins by the exogenously applied dual-epitope linker peptide GFP-SYN. Using this strategy, we demonstrated retrograde VSR recycling to the cis-Golgi as being the most distant compartment upstream of the TGN/EE. Together with the fact that VSRs reload ligands after recycling, this supports the concept of bidirectional VSR transport.

On the basis of our investigations, we now present the following concept for the operation of VSR-mediated vacuolar sorting in the plant endomembrane system (Fig. 6). Newly synthesized VSRs bind ligands in the early secretory pathway[23,27,41,42] at neutral pH[21,26,43] and transport them to the TGN/EE, where they ultimately release their ligands[27], due to a shift in compartmental pH. The TGN/EE is the most acidic compartment en route to the vacuole[26,43,44], since it harbors characteristic V-ATPases[28] that are absent from the MVBs/LEs[45], thus preventing further acidification. Therefore, the locations for binding and release of ligands are in agreement with the initially recorded pH dependency for VSR-

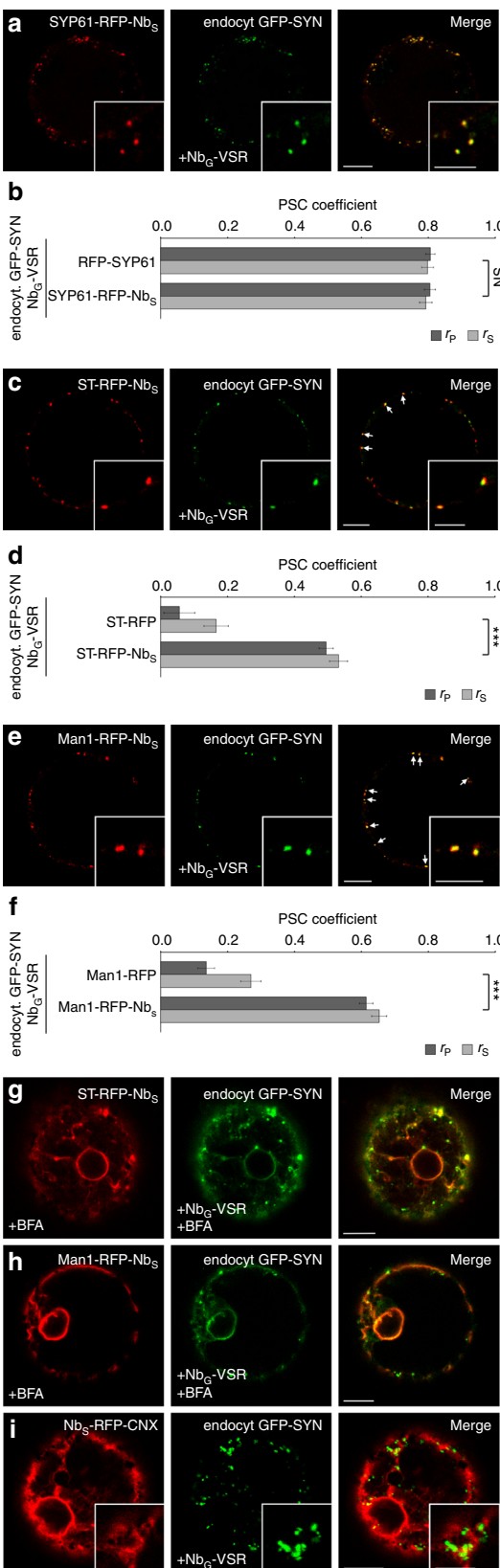

**Fig. 4** The cis-Golgi stack is the target of VSR recycling. GFP-SYN-labeled Nb$_G$-VSR is locked to the anchors in **a** the TGN/EE (SYP61-RFP-Nb$_S$), and after recycling to **c** trans-Golgi (ST-RFP-Nb$_S$) and **e** cis-Golgi (Man1-RFP-Nb$_S$) anchors but does not reach **i** the ER anchor Nb$_S$-RFP-CNX. PSC coefficients of the labeled Nb$_G$-VSR with **b** the marker Syp61-RFP or the anchor Syp61-RFP-Nb$_S$, presented/calculated as in Fig. 2g with $n = 10$, $P \geq$ 0.05, NS not significant, **d** the marker ST-RFP or the anchor ST-RFP-Nb$_S$, presented/calculated as above with $n = 10$, ***$P < 0.001$ and **f** the marker Man1-RFP or the anchor Man1-RFP-Nb$_S$, presented/calculated as in **d**. Graphs show a representative sample of two independent experiments. **g**, **h** BFA-treatment of samples from **c**, **e** for 1 h at 20 μM triggers fusion of Golgi with ER, verifying Golgi localization of locked VSRs from **c**, **e**. Scale bars 10 μm, insets 5 μm, showing magnifications

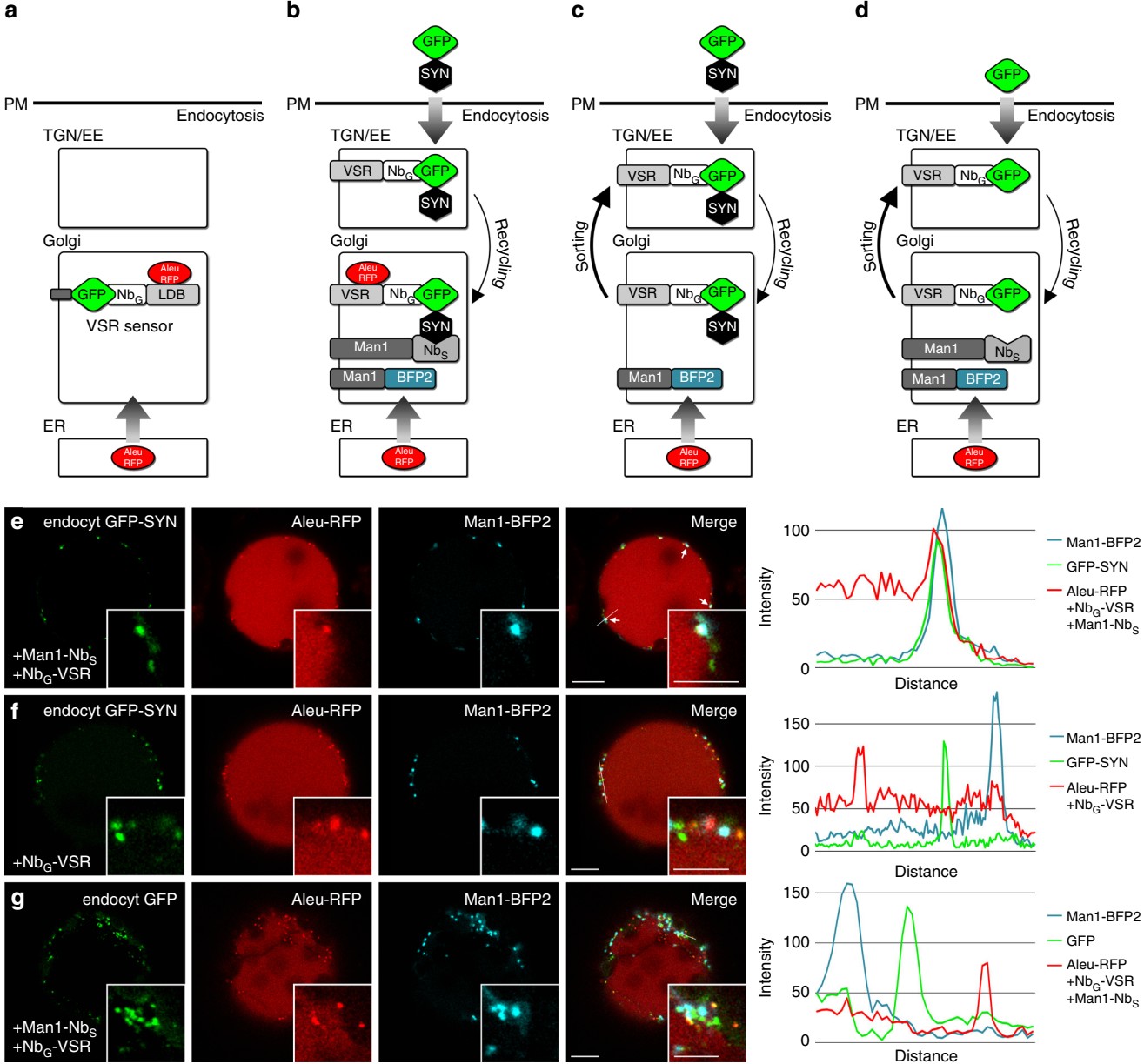

**Fig. 5** VSRs bind ligands after recycling. **a** Targeted VSR sensors (NbG-LDB) were shown to bind Aleu-RFP ligands in the Golgi[27] **b** GFP-SYN-labeled NbG-VSRs are locked to the anchor Man1-NbS in the *cis*-Golgi, positively identified by the marker Man1-BFP2. Ligand-binding of recycled full-length VSRs is assessed by colocalization with ligands (Aleu-RFP). Controls with cycling VSRs that lack **c** the anchor or **d** the SYN epitope at the labeling GFP for the VSR lockdown, result in VSR localization at the TGN/EE, which does not promote ligand binding. **e** GFP-SYN-labeled NbG-VSRs are locked after recycling in the *cis*-Golgi and colocalize with the Golgi marker Man1-BFP2 and bind the ligand Aleu-RFP, as shown by the overlapping signal peaks in the line intensity plot (see **b**). **f**, **g** Not locked VSRs (see **c** and **d**) do not localize to the Golgi and thus do not bind the ligand Aleu-RFP as judged by the separated peaks in the line intensity plots. Scale bars 10 μm, insets 5 μm, showing magnifications

ligand interactions in vitro[21]. After release in the TGN/EE, ligands progress without further involvement of VSRs onwards to the lytic vacuole by default[27]. This occurs due to a maturation event of the TGN/EE that results in the biogenesis of a MVB/LE[46,47]. While fusion of the MVB/LE with the vacuole represents the final step in the vacuolar delivery of ligands[46] it is unrelated to VSR function. VSRs, however, recycle from the TGN/EE back to the *cis*-Golgi, for ligand reloading and renewed rounds of ligand delivery to the TGN/EE. Considering the lifespan of VSRs greatly exceeding the time it takes for a round of transport, it is plausible to assume that cycling VSRs bear the brunt of the ligand transport from the Golgi

to the TGN/EE with only a minor contribution of de novo synthesized VSRs, binding their ligands in the ER.

## Methods

**Plant materials**. *Nicotiana tabacum* L. SR1 was grown on Murashige and Skoog medium supplemented with 2 % (w/v) sucrose, 0.5 g L$^{-1}$ MES and 0.8 % (w/v) Agar at pH 5.7 in 16/8 h light–dark cycles at 22 °C.

**Plasmid constructs**. All constructs are given in Supplementary Table 1. DNA manipulations were performed according to established procedures, using pGreen-nII[48]-based vectors and *Escherichia coli* MC1061. An anti-SYN nanobody sequence was generated by reverse translation of the amino acid sequence NbSyn87 without

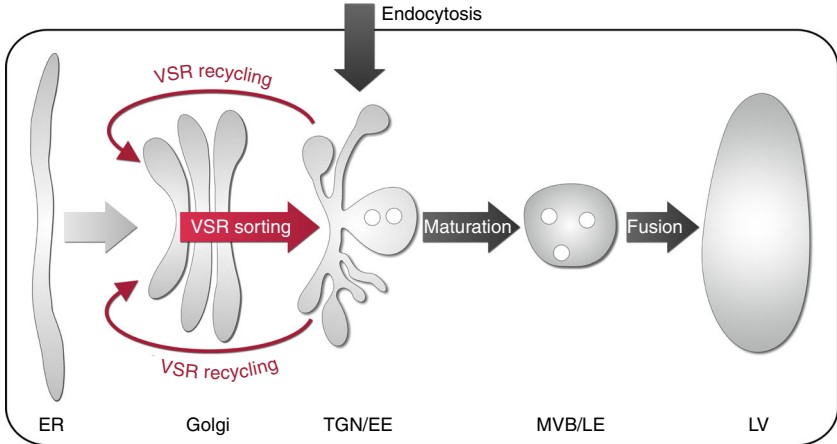

**Fig. 6** Model for receptor-mediated vacuolar sorting in plants. VSRs bind ligands in the early secretory pathway and transport them to the TGN/EE. There, the ligands are released from the VSR. Next, VSRs are recycled back to the *cis*-Golgi stack for further rounds of ligand transport. Post-TGN/EE transport of released vacuolar cargo ligands but also endocytosed proteins occurs independent of VSRs and travel to the lytic vacuole per default. Transport in this route is mediated by multivesicular bodies, the late endosomes (MVBs/LEs). They bud off the TGN/EE in a maturation-based step and confer cargo delivery by their ultimate fusion with the lytic vacuole (LV)

the C-terminal 6xHis tag[32], optimized for *Arabidopsis*-specific codon usage (EMBOSS Backtranseq), modified with an N-terminal HA-tag and chemically synthesized (GeneArt Gene Synthesis). The blue fluorescent protein mTagBFP2 (GenBank AIQ82697.1) was also generated by reverse translation of the amino acid sequence, optimized for *Arabidopsis*-specific codon usage (EMBOSS Backtranseq) and chemically synthesized (GeneArt Gene Synthesis).

All VSR constructs were assembled from AtVSR4 (GenBank accession no. NM_127036) and fused to the GFP nanobody[27]. The red fluorescent compartment-specific anchors carry a monomeric RFP[48]. The correct localization of all generated VSR-/marker-fluorophore fusions was verified.

**Protoplast isolation and gene expression**. Protoplasts were isolated from perforated leafs by overnight incubation in incubation solution (3.05 g L$^{-1}$ Gamborg B5 Medium, 500 mg L$^{-1}$ MES, 750 mg L$^{-1}$ CaCl$_2$·2H$_2$O, 250 mg L$^{-1}$ NH$_4$NO$_3$ adjusted to pH 5.7 with KOH) supplemented with 0.2 % w/v macerozyme and 0.4 % w/v cellulase) at 25 °C in the dark. They were rebuffered by washing them three times in 50 mL electrotransfection-buffer (137 g L$^{-1}$ sucrose, 2.4 g L$^{-1}$ HEPES, 6 g L$^{-1}$ KCl, 600 mg L$^{-1}$ CaCl$_2$·2H$_2$O adjusted to pH 7.2 with KOH). 150 µL protoplasts in a total volume of 600 µL electrotransfection-buffer were electrotransfected with 1–10 µg plasmid DNA using the square-wave pulse generator EPI-2500 (Fischer) applying a pulse at 130 V for 10 ms. After transfection, each sample was supplemented with 2 ml incubation buffer and incubated for 18–24 h at 25 °C in the dark.

**Biosynthesis of fluorescent reporters**. Protoplast-secreted reporters (GFP/GFP-SYN) for endocytic uptake experiments were obtained from cell-free culture medium after expression, harvesting, sonication and clearance, ruling out contaminations with reporter-synthesizing cells during uptake experiments. For the endocytic uptake, populations of protoplasts expressing Nb$_G$/Nb$_S$-tagged constructs were supplemented with cleared reporter-containing medium for 20 h.

**Confocal microscopy and statistical analysis**. Image acquisition was performed using a Leica TCS-SP8 confocal laser-scanning microscope, equipped with a ×63 (1.2 numerical aperture) water immersion objective. Fluorophores were excited (ex) and emission (em) was detected in sequential line scanning mode using HyD detectors: mTagBFP2 (ex/em, 405 nm/407–452 nm), GFP (ex/em, 488 nm/496–525 nm) and RFP (ex/em, 561 nm/569–636 nm). Pinholes were adjusted to 1 Airy unit for each wavelength. Post-acquisition image processing and assembly of figures was performed using Adobe Photoshop CS3 and CorelDraw X8.

The linear Pearson's correlation coefficient ($r_P$) and the nonlinear Spearman's rank coefficient ($r_S$) of green and red fluorescent signals was calculated with the PSC colocalization plug-in (http://www.cpib.ac.uk/~afrench/coloc.html) for ImageJ[48]. The threshold levels were set to 10. For the statistics, 10 individual cells were analyzed and the correlation coefficients are shown as mean values with standard errors of the mean. Statistical significance was calculated with R using an unpaired, two tailed *t*-test[49].

**Analysis of the SYN nanobody–epitope interaction**. Cell-free culture medium was harvested after flotation of electrotransfected tobacco protoplasts for 5 min at 80 g in sealed pre-punctured tubes, using insulin syringes. Afterwards, cells were harvested by addition of 7.5 mL of 250 mM NaCl, sedimentation for 7 min at 80 g, followed by removal of the supernatant. The culture medium was cleared by centrifugation at 20,000×*g* for 15 min at 4 °C and diluted with α-amylase extraction buffer (50 mM acid malic, 50 mM sodium chloride, 2 mM calcium chloride, 0.02% (w/v) sodium azide). Cell samples were extracted in a total volume of 250 µg with α-amylase extraction buffer, sonicated and centrifuged at 20,000 g for 15 min at 4 °C. The supernatant was recovered and employed for the reporter assay and SDS–PAGE/western blot (SDS–PAGE/WB).

The quantitative reporter transport analysis was performed in samples from the cell extracts and the culture medium, using the α-amylase reagent kit (Megazyme R-CAAR4). Individual enzymatic assays were started by addition of 30 µl of substrate solution to 30 µl of extracted and diluted sample. After incubation at 40 °C, the reaction was stopped by the addition of 150 µL of 1% w/v Trizma base. 200 µL of the reaction was transferred into a well of a microtitre plate to measure absorbance at 405 nm[50].

For SDS–PAGE/WB, samples were mixed 1:1 with freshly prepared 2× Xtreme loading dye (900 µL of sample buffer (0.1 % (w/v) bromophenol blue, 5 mM EDTA, 200 mM Tris-HCl, pH 8.8, 1 M sucrose) supplemented with 300 µL 10 % w/v SDS and 20 µL of 1 M DTT), incubated for 5 min at 95 °C and loaded onto 10 % (w/v) SDS-polyacrylamide gels. After electrophoretic separation at 40 mA, proteins were electroblotted onto nitrocellulose membranes at 200 mA. For immunodetection, membranes were incubated in blocking solution (TBS-T (6.06 g L$^{-1}$ Trizma base, 8.88 g L$^{-1}$ NaCl, 0.05 % (v/v) Tween-20), supplemented with 5 % (w/v) BSA) for 30 min and then probed with the following antibodies diluted in blocking solution: rabbit polyclonal anti-GFP (Life Technologies A6455, 1:10,000), rat monoclonal anti-RFP (ChromoTek 5F8, 1:1000) and rat monoclonal anti-HA–Peroxidase (Roche 12013819001, 1:2500). Uncropped immunoblots are given in Supplementary Fig. 7.

**Immunoprecipitation**. For anchor-epitope and VSR-epitope interaction anchors/VSRs were expressed in vivo and extracted 1:1 in 2× binding buffer (40 mM HEPES, 300 mM NaCl, 2 mM CaCl$_2$, 2 mM MgCl$_2$, pH 7.1) with 2% (v/v) CHAPS[27]. Immunoprecipitation was performed for 1 h with RFP-Trap_MA (ChromoTek, rxns-20) for the anchors and with Pierce Anti-HA Magnetic Beads (Life Technologies, 88836) for the VSRs at 4 °C. Beads were washed three times with binding buffer containing 0.4% (v/v) CHAPS and afterwards incubated with GFP-SYN/GFP, which were in parallel samples transiently expressed and recovered from the medium, overnight at 4 °C. SDS–PAGE/WB was performed as described above.

**Data availability**. The authors declare that all data supporting the findings of this study are available within the manuscript and its supplementary files or are available from the corresponding author upon request.

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

## Acknowledgements

We thank Diana Vranjkovic for technical help. The financial support of the Deutsche Forschungsgemeinschaft (PI 769/1-2 and the Collaborative Research Centre SFB 1101 "Molecular Encoding of Specificity in Plant Processes" and TP A03) and the Deutscher Akademischer Austauschdienst (Project 57057314 & 57219822) and the Southern University of Science and Technology (SUSTech)/SUSTech-PKU Institute of Plant and Food Science (IPFS) Start-up fund is gratefully acknowledged. P.P. is indebted to L. Kilmister for inspirations and support.

## Author contributions

S.F., F.F., Ü.K., and P.P. designed and analyzed experiments. S.F. performed experiments. S.F. and P.P. wrote the manuscript, P.P. conceived the study.

## Additional information

**Competing interests:** The authors declare no competing financial interests.

