## [Peer Review File · Nature Communications]

Reviewers' comments:

Reviewer #1 (Remarks to the Author):

In this manuscript, the authors investigated the recycling of VSR in plant endomembrane system. By using a novel nanobody-based labeling method, they concluded that VSR recycled from TGN/EE to cis-Golgi for ligand reloading. This is a new concept for vacuolar protein sorting in plants. The experimental design was well organized and the results were convincing under the given condition. However, I have some questions that should be addressed before publication.

1. BP-80 is known as a VSR homolog in pea. While NbG-RFP-BP80 was used as a MVB/LE marker in Fig.1d, NbG-VSR was shown to localize to the TGN/EE in Fig.2e. It was confusing to me why two VSR homologs showed different localization. Is there any difference between these two molecules or constructs? The authors should explain the reason for this issue. In addition, I'm wondering if the localization of NbG-RFP-VSR in Fig 2a is different from that of NbG-RFP-BP80 in Fig1d?

2. Given the fact that BP80 is a VSR homolog found in MVB/LE, it is very likely that NbG-RFP-BP80 recycles between MVB/LE and TGN/EE. To check this possibility, the authors should add the result of co-expression experiment of NbG-RFP-BP80 and Sec-GFP in Fig. 1.

3. In this paper, compartment markers are supposed to steadily localize to their respective compartments without recycling. In such scenario, VSR was suggested to meet compartment markers, Man1 and ST, by VSR's recycling from the TGN/EE to the Golgi stack. However, another possibility is that the compartment markers themselves recycle between compartments. Do you have any evidence that which molecules, VSR or compartment markers, or both, are actually moving along endomembrane system in this experimental condition? Have you ever checked the recycling of compartment markers? For example, what will happen if you express both SYP61-RFP-NbG and Man1-RFP-NbS together with endocyt GFP-SYN?

4. Together with their previous paper, the authors concluded that the ligands vacuolar proteins are released from VSR in the TGN/EE. If this is true, vacuolar proteins, secretory proteins, and endocytosed proteins are all co-localized in the TGN/EE. My question is how these proteins in the same TGN/EE are sorted to their destinations. The authors should explain possible mechanism in the Discussion section. I think that TGN/EE subdomains may be related to this issue. In this context, the authors should add the co-expression experiment of SYP61-RFP-NbG and Sec-GFP in Fig. 1.

5. In this paper, all the localization analysis was carried out by over-expressing target genes in tobacco protoplasts. Although this is a useful and convenient method, in my opinion, it is an artificial and stressful condition. Nowadays, we have to carefully determine the localization of our proteins under native-like condition, for example using authentic promoters with stable transformants. The authors are highly recommended to briefly touch on this issue. The authors are also expected to apply this smart nanobody technique to more natural condition in the near future.

Reviewer #2 (Remarks to the Author):

The authors try to solve the mechanism of vacuolar sorting and try to understand a number of critical questions on how the vacuolar sorption receptor after being translated is migrating from ER to golgi to transgolgi network or early endosomes and back to golgi. These questions are answered by making use in an intelligent manner of nanobodies. These nanobodies to the GFP antigen or to synuclein peptide tag are linked to various proteins, including VSR (that can also be labelled permanently with monomeric RFP) and compartment markers.

It has been shown that the expression of these constructs are not perturbing the membrane trafficking.

There might be an issue of Nbs being expressed intracellularly as intrabodies) in the sense that such intrabodies are normally easily degraded (certainly in absence of their antigen). However, in case the expression occurs in ER, Golgi or TGN then the oxidising environment of these compartments might allow the immunoglobulin disulphide bonds to be made and therefore the Nbs will be more stable.

This paper solves a number of controversies on the VSR recycling and sorting and this knowledge will be of general interest of biochemist and cell biologists.

Reviewer #3 (Remarks to the Author):

This paper is a follow on from the author's recent Nature Plants paper employing nanobody-based constructs. The conclusion of this previous work was that vacuolar sorting receptors bind cargo in the ER and are required for transport the ER and the Golgi to the TGN/EE, but after that transport to the vacuole is by a constitutive transport pathway. Here they take the technology a little further by using nanobody-epitope pairs for identifying the location of fluorescent receptors and for locking receptor constructs in specific compartments of the secretory pathway. Although in places the manuscript is a little hard to follow due to the complexity of the system the authors have devised, in general I find this to be a very nice study with well-conceived controls which demonstrates for the first time the VSRs recycle from the trans-Golgi network to the cis-Golgi.

I do have one suggestion for some extra supplementary data. Most of the results use single markers for the different compartments. Especially, the work depends on the cis-Golgi marker mannosidase 1 being spatially distinct from the trans-marker ST. The location and spread of these markers often depends on the level of expression in cells, so a figure showing lack of colocalisation of the two markers would be useful.

A minor detail – Figure 3 needs a label for panel E.

Detailed comments and answers to the reviewers' questions

Reviewer 1

In this manuscript, the authors investigated the recycling of VSR in plant endomembrane system. By using a novel nanobody-based labeling method, they concluded that VSR recycled from TGN/EE to cis-Golgi for ligand reloading. This is a new concept for vacuolar protein sorting in plants. The experimental design was well organized and the results were convincing under the given condition. However, I have some questions that should be addressed before publication.

Comment: We thank the reviewer for the very positive response to our work.

1. BP-80 is known as a VSR homolog in pea. While Nb_G-RFP-BP80 was used as a MVB/LE marker in Fig.1d, Nb_G-VSR was shown to localize to the TGN/EE in Fig.2e. It was confusing to me why two VSR homologs showed different localization. Is there any difference between these two molecules or constructs? The authors should explain the reason for this issue. In addition, I'm wondering if the localization of Nb_G-RFP-VSR in Fig 2a is different from that of Nb_G-RFP-BP80 in Fig1d?

Answer to 1:

We apologize for having caused confusion at this point. Nb_G-RFP-BP80 and Nb_G-RFP-VSR are entirely different molecules. Nb_G-RFP-VSR, which was used to test for post-translational labeling, and most importantly Nb_G-VSR, which was used for the VSR analysis are full length VSRs from *Arabidopsis*. In sharp contrast, the established MVB/LE marker Nb_G-RFP-BP80, which was used in Fig. 1d, is an extremely truncated and non-functional derivative of the sorting receptor BP80 from *Pisum sativum*. Compared to the wild-type receptor, the marker Nb_G-RFP-BP80 consists of only 71 amino acids out of BP80's total 628 amino acids. These remaining 11% represent BP80's transmembrane domain and cytosolic tail, whilst 89% of BP80, namely the entire 556 amino acid comprising luminal ligand binding domain (LBD), is missing. To avoid further confusion, we have now renamed the MVB/LE marker to Nb_G-RFP-BP80ΔLBD in the text and we have also revised the introduction of the marker proteins to also accommodate a new Supplementary Figure 1, showing colocalization of punctate marker proteins in relation to each other, as was suggested by Reviewer 3. Furthermore, we refer strictly and concisely to either molecule as "marker" or "VSR" throughout the manuscript.

It must be clear that it is not possible to colocalize the two RFP fusion proteins Nb_G-RFP-VSR and Nb_G-RFP-BP80ΔLBD directly. At the very most, the data presented in Fig. 2 show that the GFP-labeled functional full length VSR colocalizes with the TGN/EE marker RFP-SYP61 at the TGN/EE, whilst it does not colocalize with Aleu-RFP, a marker for vacuole and MVB/LE.

At this point it should be stressed that an established MVB/LE marker like GFP-BP80ΔLBD does not colocalize with the TGN/EE marker RFP-SYP61 (Confidential Figure 1).

Confidential Figure 1. Differential localization of the TGN/EE marker RFP-SYP61 and the MVB/LE marker GFP-BP80ΔLBD. Coexpression of respective markers in tobacco protoplasts. Scale bars 10μm, insets 5μm, showing magnifications.

However, due to the immediate formation of nanobody-epitope complexes in coexpression experiments (see below) it is also not possible to directly compare the localizations of Nb_G-RFP-

VSR and GFP-BP80 Δ LBD in a co-expression experiment. Nevertheless, to clearly demonstrate the different locations for the marker Nb_G-RFP-BP80 Δ LBD and the fluorescent full length VSR Nb_G-RFP-VSR, we have coexpressed each of these proteins with the TGN/EE marker GFP-SYP61 or Aleu-blue fluorescent protein (BFP)2, a soluble marker protein for the MVB/LE and the vacuole, as references. For these experiments, we had to use GFP-SYP61, an N-terminal GFP fusion of the tail-anchored SYP61 that exposes GFP to the cytosol to prevent the interaction with the nanobody in the compartmental lumen. This new analysis now clearly shows that only Nb_G-RFP-VSR colocalizes with GFP-SYP61, whilst the marker Nb_G-RFP-BP80 Δ LBD does not, and instead colocalizes, as expected, with Aleu-BFP2. We have now included this comparison between the full length VSR and the truncated marker as a new Supplementary Figure 2. The molecular mechanism for the differential localization is not clear. Considering that sorting receptors bind ligands prior to being transported forwards and release ligands before undergoing recycling, it plausible to assume that the ligand status of the receptor impinges on its transport. We therefore consider the absence of the LBD (or the loss of 89% of the total protein) to be the most likely reason for the demonstrated differential localization between the RFP-tagged or labeled functional VSR and the truncated MVB/LE marker. However, it is also evident, that the LBD of the VSR is required for its steady state localization and we have added this to the result section in the manuscript.

2. Given the fact that BP80 is a VSR homolog found in MVB/LE, it is very likely that Nb_G-RFP-BP80 recycles between MVB/LE and TGN/EE. To check this possibility, the authors should add the result of co-expression experiment of Nb_G-RFP-BP80 and Sec-GFP in Fig. 1.

Answer to 2:

As outlined above, MVB/LE markers like Nb_G-RFP-BP80 Δ LBD cannot be considered as VSR homologs and their transport features can also not be attributed to functional VSRs and vice versa. However, the analysis of putatively occurring retrograde membrane flow from the MVB/LE is certainly an interesting topic and a neutral reporter like the function-deprived MVB/LE marker might well serve as reporter for that purpose in the future. Reviewer 1 has suggested performing coexpression of the MVB/LE marker Nb_G-RFP-BP80 Δ LBD with Sec-GFP to test whether this marker recycles between the MVB/LE and the TGN/EE. The Confidential Figure 2 below, shows that Nb_G-RFP-BP80 Δ LBD and Sec-GFP strongly colocalize upon coexpression.

Confidential Figure 2. Coexpression of Nb_G-RFP-BP80 Δ LBD with secreted GFP (Sec-GFP). Coexpression of respective markers in tobacco protoplasts. Scale bars 10 μ m, insets 5 μ m, showing magnifications.

However, this is not surprising since the nanobody within Nb_G-RFP-BP80 Δ LBD was shown to bind to coexpressed GFP epitopes immediately after synthesis and folding of both components in the lumen of the ER. This, can also be seen in Fig. 1h-j, where the coexpressed Nb_G-tagged marker for the *trans*-/*cis*-Golgi and the ER act as a membrane anchor to trap the coexpressed epitope, secreted GFP. We have recently discovered this fact and utilized it to achieve nanobody-epitope interaction-triggered compartment-specific targeting of the otherwise secreted Nb_G-tagged LBD of the very same VSR to the ER, the Golgi, the TGN/EE and the MVB/LE. In these experiments,

MVB/LE targeting of the secretory Nb_G-tagged LBD was achieved by coexpression with the MVB/LE marker GFP-BP80(Δ LBD) (Künzl et al., 2016, Nature Plants 2, 16017).

Moreover, the occurrence of Nb-epitope interactions in compartments of the early secretory pathway might be regarded as an exploitable common feature of nanobody-epitope pairs, since it is also true for the tested Nb_S/SYN nanobody-epitope pair (Fig. 3b,c). There, the SYN epitope-tagged secretory α -amylase is rerouted from the early secretory pathway to the vacuolar route via the epitope binding capability of the soluble vacuolar-targeted nanobody Aleu-RFP-Nb_S. However, since nanobodies bind their epitope in coexpression experiments immediately in the ER, the first compartment of the secretory pathway, it is not possible to employ the suggested coexpression of Nb_G-RFP-BP80 Δ LBD with Sec-GFP for testing retrograde transport of Nb_G-RFP-BP80 Δ LBD from the MVB/LE. The suggested coexpression of Nb_G-RFP-BP80 Δ LBD with Sec-GFP reconstitutes a post-translationally GFP-labeled red fluorescent MVB/LE marker that is nevertheless most suitable for analyzing its anterograde transport.

[Editorial Note: Unpublished Data redacted from Peer Review File as per Authorial Request.]

3. In this paper, compartment markers are supposed to steadily localize to their respective compartments without recycling. In such scenario, VSR was suggested to meet compartment markers, Man1 and ST, by VSR's recycling from the TGN/EE to the Golgi stack. However, another possibility is that the compartment markers themselves recycle between compartments. Do you have any evidence that which molecules, VSR or compartment markers, or both, are actually moving along endomembrane system in this experimental condition? Have you ever checked the recycling of compartment markers? For example, what will happen if you express both SYP61-RFP-Nb_G and Man1-RFP-Nb_S together with endocyt GFP-SYN?

Answer to 3:

We thank the reviewer for raising this important issue. The reviewer speculates that the observed lockdown of the labeled VSR in the Golgi did not occur due to VSR recycling but occurred due to kidnaping from the TGN/EE by "cycling" *cis*- and *trans*-Golgi markers (Man1 and ST), instead. To test for such putative recycling of markers, the reviewer has asked that we "express both SYP61-RFP-Nb_G and Man1-RFP-Nb_S together with endocyt GFP-SYN". We are sorry but we find it difficult to follow this specific suggestion, since the coexpression of RFP fusion proteins for both, the Golgi and the TGN/EE would not allow for differential detection of these markers. To date, there has been no report in any eukaryotic system about the occurrence of *cis*-/*trans*-Golgi markers in post Golgi compartments. The same applies to the recycling of markers from post-Golgi compartments as a mechanism to achieve Golgi localization. However, marker proteins localize quantitatively to the Golgi and are not detectable in post-Golgi compartments like the TGN/EE. If Golgi localization were the result of retrograde recycling from the TGN/EE rather than that of lacking export, such a recycling must occur at high frequency. In such a scenario, the markers would undergo multiple rounds of transport between the TGN/EE and the Golgi stack. Here, we have demonstrated that our post-translational nanobody-based GFP labeling strategy using endocytosed GFP epitopes is both highly specific and highly efficient for the respective nanobody-epitope pair. In particular, for labeling and subsequent tracing of proteins in the TGN/EE. If nanobody-tagged Golgi markers would indeed undergo continuous cycling through the TGN/EE, as was speculated by the reviewer, the markers would be efficiently labeled with GFP due to the occurring nanobody-epitope interaction, as was demonstrated for

the VSRs. However, the opposite is true and no such GFP-labeling can be observed in the case of the Golgi markers ST-RFP-Nb_G, Man1-RFP-Nb_G and the ER marker Nb_G-RFP-CN_X (Fig. 1e-g). We have furthermore also tested if the TGN/EE anchor SYP61-RFP-Nb_G recycles to the Golgi. After post-translational labeling of the TGN/EE anchor SYP61-RFP-Nb_G with endocytosed GFP-SYN, the labeled anchor never colocalized with the Golgi marker Man1-BFP2. After BFA treatment, only signals from the Golgi marker appear in the ER and the labeled TGN/EE anchor remains in punctae, representing the TGN/EE and never appears in the ER/nuclear envelope. This furthermore demonstrates that the TGN/EE anchor does not recycle to the Golgi (new Suppl. Fig. 5).

Together [Editorial Note: Unpublished Data redacted from Peer Review File as per Authorial Request.], there is no evidence that any of the marker proteins used in this manuscript achieves their steady state distribution in the respective compartment due to a retrograde recycling step from a downstream compartment. Therefore, we have to assume that the lockdown of the VSR in the Golgi does indeed occur due to the recycling of the functional VSR from the TGN/EE, rather than due to the recycling of a marker protein. To clarify this point, we have added a short note in the manuscript, explaining that the respective marker proteins do not transit the TGN/EE in order to reach their steady state distribution.

4. Together with their previous paper, the authors concluded that the ligands vacuolar proteins are released from VSR in the TGN/EE. If this is true, vacuolar proteins, secretory proteins, and endocytosed proteins are all co-localized in the TGN/EE. My question is how these proteins in the same TGN/EE are sorted to their destinations. The authors should explain possible mechanism in the Discussion section. I think that TGN/EE subdomains may be related to this issue. In this context, the authors should add the co-expression experiment of SYP61-RFP-Nb_G and Sec-GFP in Fig. 1.

Answer to 4:

In our previous article, we presented the first direct analysis of VSR-ligand interactions in living cells. There, we employed nanobody-epitope interactions to perform a compartment-specific analysis to test for the occurrence of receptor-ligand interactions. We used a spectromicroscopical FRET-FLIM analysis, currently the most sensitive method to determine location-specific protein-protein interaction in living cells. We demonstrated that VSRs bind ligands in the ER, the *cis*- and *trans*-Golgi, and we demonstrated that they do not bind ligands in the TGN/EE or the MVB/LE, using the same compartmental markers as were used in the current study. Most important however was our demonstration that secretory proteins are delivered to the vacuole once they have reached the TGN/EE via the endocytic route. This was the first demonstration that protein transport from the TGN/EE to the vacuole does not require sorting signals at all and is thus independent of sorting receptors. This defines the vacuole as the default location for all soluble proteins that reach the TGN/EE, regardless of whether they arrive at the TGN/EE via the receptor-mediated biosynthetic route or via the endocytic route from the PM. To the best of our knowledge, no soluble protein has ever been traced from the TGN/EE to the exterior in a living cell. Together, these data and the non-discriminative post-TGN/EE transport of soluble proteins in particular, demonstrate that there is no *in vivo* evidence for the differential sorting of soluble proteins at the TGN/EE. The resolution level of VSR-ligand interactions in FRET-FLIM experiments is down to a distance of proteins with less than 10 nm for the energy transfer

to occur. In contrast to TGN/EE-specific positive controls using triggered protein interactions, the analysis of VSR-ligand analysis did not reveal any indication of VSR-ligand interactions occurring in this compartment. VSR-ligand interactions strictly depend on the chemical properties of the compartmental lumen. It is therefore hard to believe, that the entire VSR-mediated transport of ligands is based on - and achieved by - VSR-ligand interactions in TGN/EE subdomains that escaped detection entirely in these FRET-FLIM analysis using TGN/EE-anchored VSRS.

In this manuscript, we trace VSRs from the TGN/EE back to the Golgi and we demonstrate their ligand binding capability upon recycling (Fig. 5b). This triple localization shows colocalization between the dual epitope GFP-SYN-labeled full length VSR, Nb_G-VSR, its ligand Aleu-RFP and the *cis*-Golgi marker Man1-BFP2. This however occurs only if the VSR is locked down by the interaction between a SYN-nanobody-tagged Golgi marker, Man1-Nb_S, and the SYN epitope of the Nb_G-VSR-labeling GFP-SYN. If the VSR lockdown in the Golgi is compromised either by using only the single epitope label GFP (Fig. 5d) or by omitting the Nb_S-tagged Golgi anchor (Fig. 5c), the labeled VSR no longer localizes to the Golgi, but localizes to the TGN/EE, as was demonstrated already in Figure 2. However, none of these cases resulted in detectable colocalization between the TGN/EE-localizing VSR and the ligand as was seen for the Golgi-localizing VSR at the lockdown situation, indicating once again the lack of VSR-ligand interaction in this compartment.

Our data do not support the occurrence of VSR-ligand interaction in the TGN/EE and there is no published evidence for such interactions occurring in this compartment either. Together with the lack of unequivocally established markers for postulated specific subdomains of the TGN/EE, we feel that the data provided in this manuscript do not justify such a discussion about VSR-mediated ligand sorting in subdomains of the TGN/EE as was requested by the reviewer. However, in this context, the reviewer has also suggested coexpressing the TGN/EE marker SYP61-RFP-Nb_G together with Sec-GFP (Confidential Figure 4 (below)).

Confidential Figure 4. Coexpression of SYP61-RFP-Nb_G with secreted GFP (Sec-GFP). Coexpression of respective proteins in tobacco protoplasts. Scale bars 10µm, insets 5µm, showing magnifications.

For the reasons outlined in detail in our answer to point 2, above, both proteins perfectly colocalize as a result of the occurring interaction between the Nb_G of SYP61-RFP-Nb_G and its epitope, Sec-GFP, upon synthesis and folding of both proteins in the lumen of the ER. We therefore suggest omitting this experiment from the manuscript as well.

5. In this paper, all the localization analysis was carried out by over-expressing target genes in tobacco protoplasts. Although this is a useful and convenient method, in my opinion, it is an artificial and stressful condition. Nowadays, we have to carefully determine the localization of our proteins under native-like condition, for example using authentic promoters with stable transformants. The authors are highly recommended to briefly touch on this issue. The authors are also expected to apply this smart nanobody technique to more natural condition in the near future.

Answer to 5:

With all due respect we are not in agreement with the reviewer on this issue. "Native-like" conditions are very difficult to define. For instance the widely used suspension-cultured BY-2 cell system is *ex planta* and not "native". Nor for that matter are axenically-cultured *Arabidopsis* plantlets since they are never exposed to environmental stress factors as found in nature. The notion that protoplasts are stressed is a common misbelief which is not supported by the facts. Numerous publications have shown that protoplasts are not stressed (e.g. Niu and Sheen, 2012, Transient expression assays for quantifying signaling output, *Methods in Molecular Biology* 876: 195-206). Quite to the contrary, protoplasts have now become a preferred system to study stress responses (Confraria and Baena-Gonzalez, 2016, Using *Arabidopsis* Protoplasts to Study Cellular Responses to Environmental Stress, *Methods in Molecular Biology* 1398: 247-269). Protoplasts possess a fully functional vacuolar sorting and transport machinery, and no evidence has ever been provided showing that the basic biochemical properties of intracellular compartments differ between mesophyll protoplasts and cells *in situ*.

Due to our highly reproducible expression platform, protoplasts are currently the only cellular system that allows for fine-tuned coexpression of four different plasmids to achieve coordinated expression of these proteins in specific ratios to each other. Moreover, it has to be considered that all proteins used in this study were purposefully designed for use in a specific experiment. Similarly, the markers were also carefully tested for their correct localizations. It is therefore evident that none of these constructs can be expressed under the control of an "authentic" promoter. In sharp contrast to the experimental strategy presented, stable transformants do not allow any adjustment of the expression levels of coexpressed proteins, which may greatly vary between individual plants. This however also applies to all currently used stably transformed transgenic lines that express man-made-reporter or effector proteins. However, the most critical part of the work presented here is the TGN/EE-specific post-translational GFP-labeling of proteins. As explained in detail above, the sophisticated nanobody-epitope-based GFP-labeling cannot be achieved by the simple coexpression of GFP and Nb_G fusion proteins in a transgenic line.

Reviewer 2

The authors try to solve the mechanism of vacuolar sorting and try to understand a number of critical questions on how the vascular sorption receptor after being translated is migrating from ER to golgi to transgolgi network or early endosomes and back to golgi. These questions are answered by making use in an intelligent manner of nanobodies. These nanobodies to the GFP antigen or to synuclein peptide tag are linked to various proteins, including VSR (that can also be labelled permanently with monomeric RFP) and compartment markers.

It has been shown that the expression of these constructs are not perturbing the membrane trafficking.

There might be an issue of Nbs being expressed intracellularly (as intrabodies) in the sense that such intrabodies are normally easily degraded (certainly in absence of their antigen). However, in case the expression occurs in ER, Golgi or TGN then the oxidising environment of these compartments might allow the immunoglobulin disulphide bonds to be made and therefore the Nbs will be more stable.

This paper solves a number of controversies on the VSR recycling and sorting and this knowledge will be of general interest of biochemist and cell biologists.

Comment: We thank this reviewer for the very positive response to our work. We appreciate the notion about the stability of the nanobodies and we will analyze this further in detail.

Reviewer 3

This paper is a follow on from the author's recent Nature Plants paper employing nanobody-based constructs. The conclusion of this previous work was that vacuolar sorting receptors bind cargo in the ER and are required for transport the ER and the Golgi to the TGN/EE, but after that transport to the vacuole is by a constitutive transport pathway. Here they take the technology a little further by using nanobody-epitope pairs for identifying the location of fluorescent receptors and for locking receptor constructs in specific compartments of the secretory pathway. Although in places the manuscript is a little hard to follow due to the complexity of the system the authors have devised, in general I find this to be a very nice study with well-conceived controls which demonstrates for the first time the VSRs recycle from the trans-Golgi network to the cis-Golgi.

I do have one suggestion for some extra supplementary data. Most of the results use single markers for the different compartments. Especially, the work depends on the cis-Golgi marker mannosidase 1 being spatially distinct from the trans-marker ST. The location and spread of these markers often depends on the level of expression in cells, so a figure showing lack of colocalisation of the two markers would be useful.

Comment: We thank the reviewer for his appreciation of our work. We fully agree with the suggested experiment to show the relation of the marker signals to each other. We consider this to be a particularly useful suggestion and have therefore also included the marker proteins that also result a "punctate pattern" (TGN/EE & MVB/LE). The various co-expressions are shown in Supplemental Figure 1.

A minor detail – Figure 3 needs a label for panel E.

We have checked and verified the labeling of all panels in Figure 3.

REVIEWERS' COMMENTS:

Reviewer #1 (Remarks to the Author):

The authors have addressed all of my concerns. I appreciate their sincere responses. I have no further comments and recommend acceptance of the manuscript.

Reviewer #3 (Remarks to the Author):

The authors have made a comprehensive effort to address the referee's concerns about their paper. They have certainly satisfied my comments and created a new supplementary figure showing co-expressions of some of their markers. Most of the other issues have been raised by referee one and the authors have given detailed responses backed up by data. Just one issue that they disagree with the referee is that of protoplasts being an artificial stressed situation. I must say that I agree with the author's rebuttal here. Protoplasts are in many respects no more stressed than plantlets growing on agar under totally artificial conditions. There are also many other arguments concerning expression levels, native promoters etc. etc. in transformed systems. Also it can be said that as protoplasts are busily trying to build a new cell wall that the secretory pathway must be stimulated or upregulated making them a good candidate system for studying the pathway. Also they can withstand transformation with multiple constructs in different combinations which is very tricky in stable transformants. Most of the problems associated with the use of protoplasts stems from poor microscopy and this is not really the case in this paper.

ANSWERS TO REVIEWERS' COMMENTS:

Reviewer #1 (Remarks to the Author):

The authors have addressed all of my concerns. I appreciate their sincere responses. I have no further comments and recommend acceptance of the manuscript.

Answer:

We thank this reviewer for the encouraging comments and the appreciation of our work.

Reviewer #3 (Remarks to the Author):

The authors have made a comprehensive effort to address the referee's concerns about their paper. They have certainly satisfied my comments and created a new supplementary figure showing coexpressions of some of their markers. Most of the other issues have been raised by referee one and the authors have given detailed responses backed up by data. Just one issue that they disagree with the referee is that of protoplasts being an artificial stressed situation. I must say that I agree with the author's rebuttal here. Protoplasts are in many respects no more stressed than plants growing on agar under totally artificial conditions. There are also many other arguments concerning expression levels, native promoters etc. etc. in transformed systems. Also it can be said that as protoplasts are busily trying to build a new cell wall that the secretory pathway must be stimulated or upregulated making them a good candidate system for studying the pathway. Also they can withstand transformation with multiple constructs in different combinations which is very tricky in stable transformants. Most of the problems associated with the use of protoplasts stems from poor microscopy and this is not really the case in this paper.

Answer: We thank this reviewer for the supportive positive comments and the constructive feedback on our work. We most sincerely appreciate the comments on the suitability of protoplasts for the analysis of protein sorting and transport processes in vivo.